# Estimating the Risk of Depression and Care Burden among Dementia Caregivers: A Feasibility Study

Omid Ghadami, Kruthika Gaddam, Mahesh Moodukonaje, *Hyeju Jang, *Hee-Tae Jung

*Luddy School of Informatics, Computing, and Engineering*
*Indiana University Indianapolis, Indianapolis, USA*
{hyejuj,heetjung}@iu.edu
*Co-corresponding Authors

*Abstract*—Caregivers of people living with dementia (PwD) are highly susceptible to depression due to the substantial care burden they experience. While caregivers often neglect their own mental health and rarely seek necessary medical services, they often communicate their perceived burdens and depressive symptoms to social workers, who serve as critical points of contact for their loved ones. Thus, accurately estimating the risk of depression and care burden through these conversational interactions may facilitate early screening and intervention. This feasibility study explored the effectiveness of using caregivers' demographic information and their narrative descriptions of caregiving experiences to estimate depression risk and caregiver burden. Utilizing Natural Language Processing (NLP) and machine learning (ML) techniques, we trained estimation models based on data from 65 caregivers, using clinical screening measures—the Patient Health Questionnaire-8 (PHQ-8) and Zarit Burden Interview (ZBI)—as reference standards. The best-performing models achieved F1 Scores of 0.74 and 0.78 for depression and burden estimation, respectively. These results demonstrate the potential of leveraging demographic and conversational data to enable early identification of caregiver distress, facilitating timely interventions that could ultimately enhance caregiver well-being and improve the quality of care provided to PwD.

*Index Terms*—Early intervention, Depression, Caregiver burden, Dementia caregivers, Risk estimation, Machine Learning (ML), Natural Language Processing (NLP), Patient Health Questionnaire (PHQ-8), Zarit Burden Interview (ZBI).

## I. Introduction

The global population is aging, leading to an increase in dementia cases worldwide [1]. Caring for people living with dementia (PwD) is a demanding responsibility, often placing a significant burden on caregivers [2]. Research indicates that caregivers of PwD experience higher rates of depression compared to the general population [3]–[5]. Neglecting the mental health issues of caregivers negatively impacts their perceived quality of life (QoL) [6]. This reduction in caregivers' QoL diminishes the quality of care provided, subsequently lowering the QoL of PwD [7]. Therefore, it is crucial to identify the risk of depression and caregiver burden and ensure that caregivers receive timely medical assistance.

Despite the acknowledged importance, caregivers often lack the time to seek medical advice for their own conditions due to their caregiving responsibilities and consequently are not screened for mental health issues [8]. As a result, they overlook symptoms of depression, delaying screening and

medical help [7]. While clinically validated screening tools exist [9], their adoption in practice remains limited, and more conversational approaches are deemed necessary [10]. Interestingly, social workers, who serve as the first point of contact for PwD at day activity centers or other care facilities, regularly have conversations with caregivers about their perceived burden and depressive symptoms [11]. These conversations make social workers an optimal channel for screening the risk of depression and perceived care burden. Devising accessible screening tools suitable for these settings could facilitate early detection and timely interventions for depression and perceived burden in caregivers. However, existing studies on caregiver burden and depression primarily rely on extensive survey data [12]. When narrative data have been analyzed, researchers have predominantly used unlabeled social media content (e.g., Reddit posts), analyzed via topic modeling or sentiment analysis, limiting practical utility in real-world caregiving contexts [13], [14]. Hence, there is a clear need to develop practical models for early risk estimation, leveraging conversational narratives similar to the interactions caregivers naturally have with social workers.

In this work, we demonstrate that caregivers' demographic information, easily accessible to social workers, along with conversations about caregivers' perceived challenges, can be used to estimate the risk of their depression and caregiving burden. Specifically, we propose an analytical pipeline to train models for these tasks utilizing: 1) caregivers' demographic information only (hereafter referred to as demographic feature set), 2) caregivers' informal narratives about their caregiving experiences only (language feature sets), and 3) a combination of caregivers' demographic information and narratives (combined feature sets). These approaches build upon prior research indicating that demographic factors such as age, race, and socioeconomic status correlate with increased caregiver burden and depression risk [15]. Additionally, studies suggest that narrative language patterns can reveal emotional states and psychological distress [16]. With a specific emphasis on translational impact in medical practice, we propose an analytical pipeline to estimate burden and depression using accepted clinical screening measures: the Patient Health Questionnaire-8 (PHQ-8) [9] and the Zarit Burden Interview (ZBI) [17]. To demonstrate feasibility, we collected demographic and narrative data from 65 dementia caregivers and trained estimation

models. Our results show that the proposed analytical pipeline achieves acceptable performance.

## II. METHODS

To assess the risk of depression and perceived burden among dementia caregivers, we developed classification models using traditional machine learning (ML) algorithms and performed feature importance analysis. This section outlines the complete pipeline, from data collection to model interpretation. The study protocol was approved by the Institutional Review Board of Indiana University Indianapolis (IRB #22162).

### A. Study Participants

Recruitment efforts were conducted from March through April 2024, utilizing posters in public spaces, churches, daycare centers, and online community support groups (e.g., Reddit). The inclusion criteria required participants to be 18 years old or older, have English as their primary language, reside in the United States, and serve as caregivers for PwD. Both primary and secondary caregivers were included, encompassing individuals who assisted PwD with daily activities or provided healthcare support (e.g., accompanying patients to appointments and administering medications). To the best of our knowledge, no publicly available demographic and narrative datasets labeled with clinically validated assessment scores, such as the Patient Health Questionnaire-8 (PHQ-8) and the Zarit Burden Interview (ZBI), exist. Therefore, we collected our own data to validate the proposed analytical approach. Consequently, a total of 65 caregivers of PwD were recruited for this study. TABLE I summarizes the demographic characteristics of our study participants.

### B. Data Collection

Data collection involved two steps: 1) self-administered surveys that included demographic questions, a depression screening measure (PHQ-8), and a caregiver burden assessment (ZBI); and 2) semi-structured interviews conducted via Zoom. The total scores from the PHQ-8 and ZBI surveys were computed to label participants as belonging to the depression-risk group (i.e., PHQ-8 score $\geq$ 10) and the high burden group (i.e., ZBI score $\geq$ 20), which were subsequently used as target labels for classification. During the Zoom interviews, participants were asked to turn off their video cameras to promote candid responses, enabling them to freely express their experiences without being influenced by nonverbal cues. We employed open-ended interview probes to better understand participants' symptoms of depression and perceived burden (see Appendix). These probes were adapted from open-ended questions regarding burden and depression used in prior research [18]. The audio data from the interviews were transcribed and converted into text for subsequent analysis.

### C. Feature Extraction and Selection

We used three types of features for the proposed pipeline: demographic features, language features from narratives, and combined features. For narrative data, we employed two different strategies depending on the type of model. For traditional

TABLE I
GROUPED CHARACTERISTICS OF CAREGIVERS OF PwD (N=65)

| Characteristics | Levels | n (%) |
|---|---|---|
| Age | 18-34 years | 59 (90.77) |
| | 35 years and above | 6 ( 9.23) |
| Sex | Male | 48 (73.85) |
| | Female | 17 (26.15) |
| Race | Black or African American | 55 (84.62) |
| | White | 6 ( 9.23) |
| | Other | 3 ( 4.61) |
| | Unknown | 1 ( 1.53) |
| Ethnicity | Hispanic or Latino | 1 ( 1.54) |
| | Not Hispanic or Latino | 57 (87.69) |
| | Unknown | 7 (10.77) |
| Location | Urban | 42 (64.61) |
| | Suburban | 15 (23.03) |
| | Rural | 7 (10.77) |
| | Unknown | 1 (1.54) |
| Employment | Employed full-time | 31 (47.69) |
| | Employed part-time | 31 (47.69) |
| | Other | 3 (4.62) |
| Education | Bachelors or Above | 50 (76.92) |
| | No Bachelors Degree | 15 (23.08) |
| Income | Under $25,000 | 12 (18.46) |
| | $25,000 - $50,000 | 19 (19.23) |
| | $50,001 - $75,000 | 13 (20) |
| | Over $75,000 | 20 (30.76) |
| | Unknown | 1 (1.54) |
| Health Insurance | Yes | 48 (73.85) |
| | No | 16 (24.62) |
| | Not Sure | 1 (1.54) |
| Relationship to loved one | Parent | 32 (49.23) |
| | Other family member | 22 (33.85) |
| | Sibling | 6 (9.23) |
| | Spouse/Partner | 5 (7.69) |
| Years caring | 1-2 years | 15 (23.03) |
| | 3-5 years | 39 (60) |
| | 6 years and more | 11 (16.92) |
| Daily caregiving hours | 2-4 hours | 9 (13.85) |
| | 5-8 hours | 36 (55.38) |
| | More than 9 hours | 20 (30.77) |
| Depressed | No | 39 (60.00) |
| | Yes | 26 (40.00) |
| Burden Level | No High Burden | 26 (40.00) |
| | High Burden | 39 (60.00) |

ML algorithms that require feature engineering, we applied Bag-of-Words (BoW) representations. Pretrained Language Models (PLMs) such as BERT were used in an end-to-end manner, where the model learns directly from the raw text input without manual feature engineering.

*1) Demographic Features:* We used all the characteristics obtained from the demographic survey questions (detailed in TABLE I) as categorical features. For missing values in *Race*, *Ethnicity*, *Income*, and *Location*, we assigned the value "Unknown". For traditional ML algorithms, we employed the Chi-square test for feature selection.

*2) Language Features from Narratives:* To prepare the narrative text data for analysis using traditional ML models, we applied preprocessing steps, including lowercasing, tokenization, stop word removal, and punctuation removal, using SPACY 3.7.5, a Python-based NLP toolkit [19]. After preprocessing, we constructed three BoW unigram frequency feature sets. The first of these sets, referred to as **UNI**, computed un-

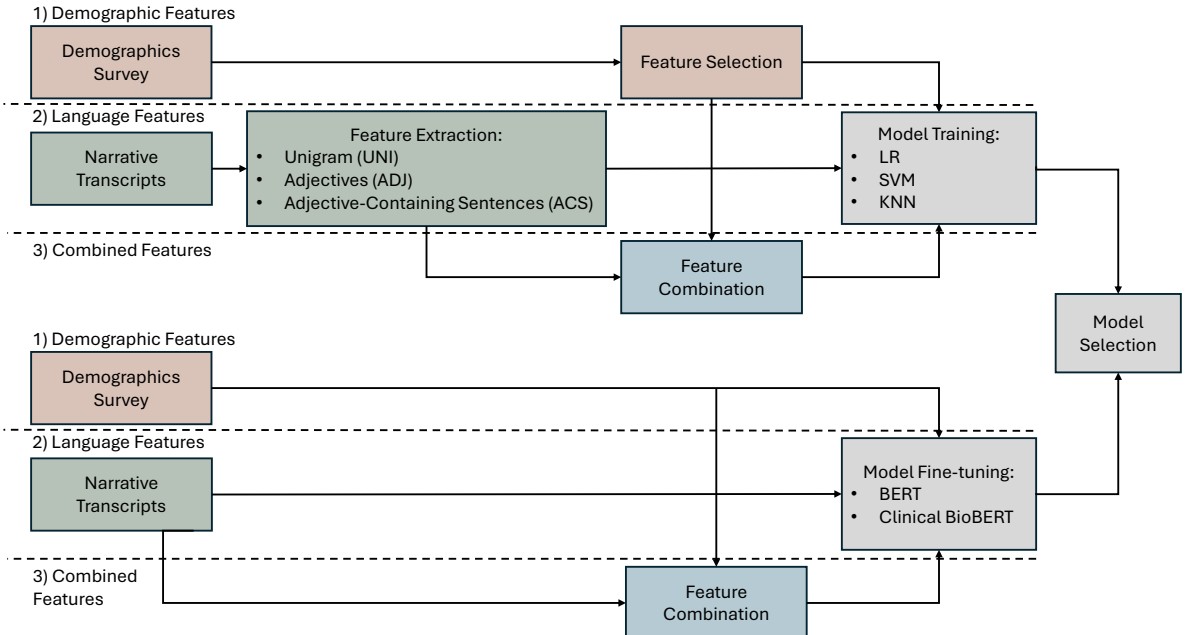

Fig. 1. The analytic pipeline used to train and fine-tune estimation models to predict the risk of depression and perceived burden among dementia caregivers. The pipeline includes traditional ML methods (Logistic Regression, SVM, and kNN) and PLMs (BERT and Clinical BERT), utilizing three feature sets: 1) demographic features only, 2) narrative language features only, and 3) a combination of demographic and narrative features.

igram frequencies across the entire narrative text, resulting in $3,867$ features. The second feature set, **ADJ**, focused on adjectives, yielding $641$ features. This approach was motivated by prior research indicating that adjectives in Western languages often convey more detailed information about the person who produced the narrative [20]. The third set, **ACS** (Adjective-Containing Sentences), calculated unigram frequencies from sentences including at least one adjective, producing $3,601$ features. Expressions of depression or caregiver burden are often closely linked to emotional and descriptive language, including adjectives [21]. We hypothesized that sentences with adjectives would provide richer contextual information, offering more focused insights into caregivers' experiences. Together, these three representations enabled a multi-layered analysis of the narrative content (Fig. 1).

*3) Combined Features:* The demographic features were combined with NLP-derived features, resulting in three feature sets: 1) Demo+UNI, 2) Demo+ADJ, and 3) Demo+ACS.

### D. Model Training and Selection

We employed six different supervised ML algorithms to identify the most effective approach and leverage diverse modeling strengths. Specifically, we included Logistic Regression (LR; a linear parametric model), Random Forest (RF; a non-parametric model), K-Nearest Neighbors (kNN; a non-parametric model), Support Vector Machine (SVM; a kernel-based model), BERT [22], and Clinical BioBERT [23] (both PLMs).

We selected general-domain BERT, which is pretrained on large-scale corpora such as BookCorpus and Wikipedia, under the assumption that caregiver narratives can be treated as general natural language text. To examine the effect of domain-specific pretraining, we also included Clinical BioBERT, which is pretrained on biomedical and clinical texts, such as the MIMIC-III dataset [24]. Given the clinical relevance of our caregiving-focused narratives, this allowed us to explore how domain alignment influences model performance in estimating caregiver depression and burden. An overview of the analytical pipeline and methods is illustrated in Fig. 1.

Feature selection, described previously, was conducted independently within each fold of the 65-fold Leave-One-Subject-Out Cross-Validation (LOSO-CV) to prevent data leakage. After feature selection, separate estimation models for depression risk and caregiver burden were trained for all feature sets and algorithms. We implemented traditional ML algorithms (LR, RF, kNN, and SVM) using scikit-learn 1.2.2 [25] with default hyperparameters. Specifically, we employed five neighbors ($k$ = 5) for the kNN classifier and a linear kernel for the SVM classifier.

For PLM-based neural models, we explored three configurations. For the demographic-only model, features were binary-encoded and fed directly into a classification layer. For the narrative-only models, the input text was processed by a PLM serving as a text encoder; the resulting contextualized representation—specifically the [CLS] token embedding, summarizing the entire input sequence—was passed to a classification layer. For the combined model we used a late fusion approach, narrative text was processed through a PLM using mean pooling, while demographic features

TABLE II
PERFORMANCE METRICS (PRECISION, RECALL, F1-SCORE) FOR TRADITIONAL CLASSIFIERS ACROSS DIFFERENT FEATURE SETS FOR DEPRESSION AND BURDEN

| Feature Set | Depression | | | | | | | | | | | | Burden | | | | | | | | | | | |
|---|---|---|---|---|---|---|---|---|---|---|---|---|---|---|---|---|---|---|---|---|---|---|---|---|
| | LR | | | SVM | | | kNN | | | RF | | | LR | | | SVM | | | kNN | | | RF | | |
| | P | R | F1 | P | R | F1 | P | R | F1 | P | R | F1 | P | R | F1 | P | R | F1 | P | R | F1 | P | R | F1 |
| Demo | 0.67 | 0.46 | 0.55 | 0.58 | 0.42 | 0.49 | **0.73** | 0.31 | 0.43 | 0.70 | 0.54 | 0.61 | 0.78 | 0.72 | 0.75 | **0.90** | 0.69 | **0.78** | 0.64 | 0.74 | 0.69 | 0.71 | 0.69 | 0.70 |
| UNI | 0.65 | 0.68 | 0.64 | 0.60 | 0.58 | 0.59 | 0.44 | 0.42 | 0.43 | 0.60 | 0.12 | 0.19 | 0.74 | 0.67 | 0.70 | 0.70 | 0.59 | 0.64 | 0.64 | 0.69 | 0.67 | 0.64 | 0.95 | 0.76 |
| ADJ | 0.48 | 0.54 | 0.51 | 0.42 | 0.47 | 0.54 | 0.43 | 0.81 | 0.56 | 0.50 | 0.23 | 0.32 | 0.62 | 0.64 | 0.63 | 0.62 | 0.67 | 0.64 | 0.60 | 0.87 | 0.71 | 0.64 | 0.87 | 0.74 |
| ACS | 0.67 | 0.63 | 0.65 | 0.64 | 0.61 | 0.65 | 0.63 | 0.48 | 0.50 | 0.40 | 0.08 | 0.13 | 0.77 | 0.69 | 0.73 | 0.74 | 0.72 | 0.73 | 0.68 | 0.67 | 0.68 | 0.60 | 0.92 | 0.73 |
| Demo+UNI | 0.65 | 0.65 | 0.65 | 0.63 | 0.65 | 0.64 | 0.42 | 0.38 | 0.40 | 0.57 | 0.15 | 0.24 | 0.73 | 0.69 | 0.71 | 0.69 | 0.64 | 0.67 | 0.62 | 0.64 | 0.63 | 0.60 | 0.95 | 0.73 |
| Demo+ADJ | 0.54 | 0.58 | 0.56 | 0.54 | 0.58 | 0.56 | 0.48 | 0.58 | 0.53 | 0.50 | 0.19 | 0.28 | 0.73 | 0.77 | 0.75 | 0.70 | 0.72 | 0.71 | 0.78 | 0.72 | 0.75 | 0.65 | 0.95 | 0.77 |
| Demo+ACS | 0.68 | 0.73 | 0.70 | 0.68 | 0.81 | **0.74** | 0.46 | 0.46 | 0.46 | 0.33 | 0.08 | 0.12 | 0.72 | 0.74 | 0.73 | 0.72 | 0.74 | 0.73 | 0.72 | 0.67 | 0.69 | 0.60 | **0.97** | 0.75 |

TABLE III
PERFORMANCE METRICS (PRECISION, RECALL, F1-SCORE) FOR FINE-TUNED PLMS USING DIFFERENT FEATURE SETS

| Feature Set | Depression | | | | | | Burden | | | | | |
|---|---|---|---|---|---|---|---|---|---|---|---|---|
| | BERT | | | Clinical BioBERT | | | BERT | | | Clinical BioBERT | | |
| | P | R | F1 | P | R | F1 | P | R | F1 | P | R | F1 |
| Demo | 0.50 | 0.81 | 0.62 | 0.50 | 0.81 | 0.62 | 0.68 | 0.77 | 0.72 | 0.68 | 0.77 | 0.72 |
| Narrative | 0.61 | 0.42 | 0.50 | 0.52 | 0.50 | 0.51 | 0.70 | 0.72 | 0.71 | 0.53 | 0.62 | 0.57 |
| Demo+Narrative | 0.62 | **0.90** | 0.73 | 0.57 | 0.77 | 0.66 | 0.55 | 0.81 | 0.66 | 0.69 | 0.85 | 0.76 |

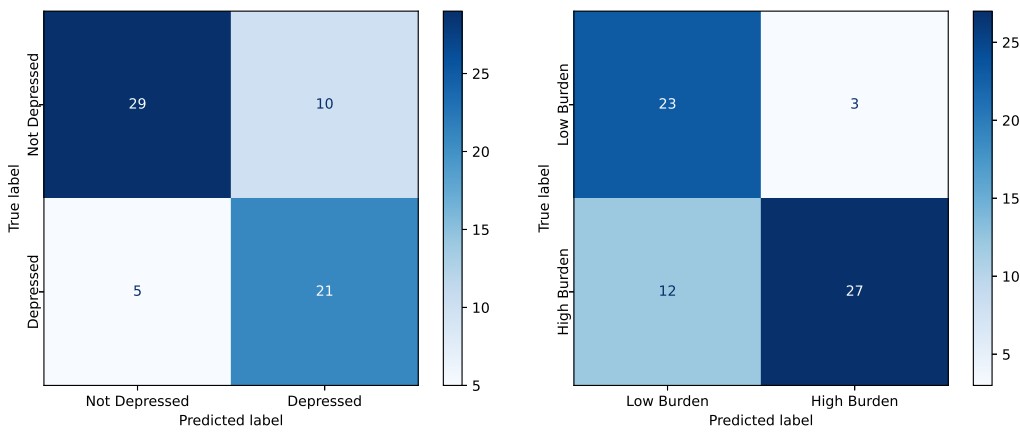

(a) Demo + ACS × SVM to estimate the risk of depression  (b) Demo × SVM to estimate the risk of burden

Fig. 2. Confusion matrices for the best-performing models to estimate the risk of depression and perceived burden

were handled by a Multilayer Perceptron (MLP) to processes the binary-encoded demographic vector. The final predictions were obtained through weighted combination of both model outputs, with fusion weights optimized via grid search. This representation, capturing both what caregivers say and who they are, was then passed to a classification layer, allowing the model to jointly leverage textual and demographic signals for prediction. Fine-tuning was implemented using the Hugging Face Transformers library [26] with PyTorch version 2.6.0 [27] and default hyperparameters.

Model performance was evaluated using Precision, Recall, and F1-score across all LOSO-CV folds. For PLMs, each fold was fine-tuned for ten epochs based on empirical observations to minimize overfitting and underfitting. The best-performing models for estimating depression and caregiver burden were identified based on the highest F1-score, as it provides a single, balanced measure that combines both precision and recall, making it more informative than either metric alone, especially in imbalanced classification tasks.

*E. Feature Importance Analysis*

To interpret the contributions of individual features to model predictions, we employed SHapley Additive exPlanations (SHAP), which assigns each feature an importance value for a particular prediction [28]. Specifically, we calculated the mean absolute SHAP values across all 65 LOSO-CV folds to assess the relative impact of each feature on the estimation of depression risk and caregiver burden.

## III. RESULTS

We trained a total of 34 models across various feature sets and algorithms to estimate depression risk and caregiver

TABLE IV
THE NUMBER OF SELECTED DEMOGRAPHIC FEATURES FOR EACH TASK.
FEATURES NOT SELECTED ARE OMITTED.

|  | Depression | Burden |
|---|---|---|
| Location | 65 | 65 |
| Years Caring | 65 | 65 |
| Income | 65 | 65 |
| Age | 65 | 1 |
| Education | 65 | 0 |
| Ethnicity | 60 | 65 |
| Race | 26 | 65 |
| Relationship to Loved One | 4 | 6 |
| Daily Caregiving Hours | 0 | 65 |

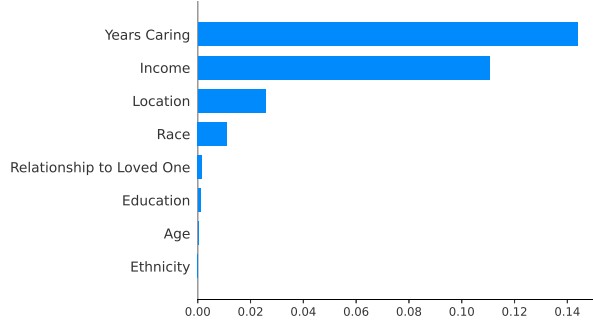

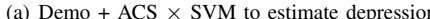

(a) Demo + ACS × SVM to estimate depression

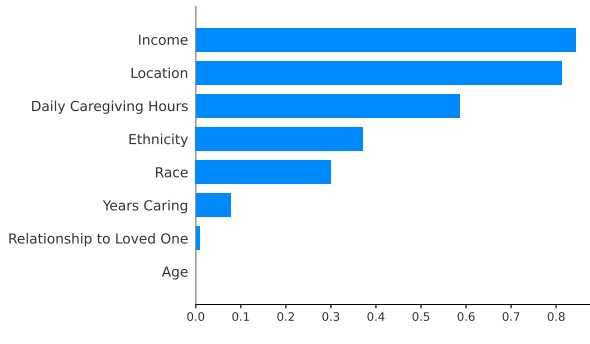

(b) Demo × SVM to estimate burden

Fig. 3. Mean absolute SHAP values for the best-performing models to estimate the risk of depression and perceived burden

burden. These models included traditional classifiers (LR, FR, SVM, and $k$NN) across multiple feature set combinations, as well as fine-tuned Transformer models (BERT and Clinical BioBERT).

*A. Model Performance*

Model performance, evaluated using Precision, Recall, and F1-score under LOSO-CV, is summarized in TABLE II for traditional models and TABLE III for PLMs. Overall, SVM demonstrated the highest performance for both estimation tasks. Specifically, based on F1-score, the best-performing depression estimation model utilized SVM with combined demographic and adjective-containing sentence features (Demo+ACS), achieving an F1-score of $0.74$. Also, the best-performing burden estimation model used SVM with demographic features alone, achieving an F1-score of $0.78$. These SVM models outperformed other traditional classifiers and the fine-tuned Transformer models in F1-score metric. In terms of other performance metrics, Random Forest using the Demo+ACS feature set achieved the highest recall ($0.97$) for the caregiver burden estimation task. For precision, the SVM model using only demographic features outperformed others with a precision of $0.90$. For depression estimation, BERT achieved the highest recall ($90$). The highest precision ($0.73$) in depression estimation was obtained by the $k$NN model using demographic feature set. Additionally, combining narrative and demographic features generally enhanced depression prediction performance (e.g., SVM Demo+ACS $0.74$ vs. SVM ACS $0.65$ F1-score).

Confusion matrices for the best-performing SVM models (Fig.2) demonstrated robust overall accuracy. However, the burden classification model misclassified 12 high-burden cases as "Low Burden," indicating difficulty in correctly identifying some positive cases (Fig.2(b)).

*B. Feature Importance*

TABLE IV summarizes how frequently each demographic feature was selected across 65 iterations for the depression and burden classification tasks. The *Location*, *Years Caring*, and *Income* feature were selected in every iteration (65/65) for both tasks, highlighting its predictive value. In contrast, features such as *Sex*, *Employment*, and *Health Insurance* were never selected, suggesting negligible predictive relevance. Several features showed task-specific relevance, including *Education*, *Age*, *Race*, and *Daily Caregiving Hours*. These results suggest depression and burden risks may be shaped by different demographic factors

Feature contributions, assessed using mean absolute SHAP values for the best models, identified *Years Caring* and *Income* emerged as the top contributors for estimation with the best depression model (Fig.3(a)), whereas *Location*, *Race*, *Relationship to Loved One*, *Education*, *Age*, and *Ethnicity* had less influence. *Income* and *Location* were the most influential demographic predictors for caregiver burden, followed by *Daily Caregiving Hours*, *Ethnicity*, and *Race* (Fig.3(b)). In contrast, *Years Caring*, *Relationship to Loved One*, and *Age* contributed less. Overall, *Income* consistently demonstrated significant predictive importance among the selected features for both estimation tasks, while *Age* and *Relationship to Loved One* had minimal impact.

## IV. DISCUSSION

Our results demonstrate the feasibility of estimating the risk of caregiver depression and burden using demographic and narrative data. Among the models evaluated, SVM yielded best-performing models, highlighting its effectiveness with limited data ($N = 65$). The lower performance of Transformer-based models (BERT and Clinical BioBERT) likely resulted from dataset constraints. Notably, Clinical BioBERT did not outperform general-domain BERT in some

cases, suggesting our narratives, though caregiving-focused, may not have contained specialized clinical language advantageous to the domain-specific model. This may be due to the nature of the interview transcripts, which include general vocabulary and conversational language. While specific to the caregiving context, the language used aligns more closely with everyday discourse than with specialized biomedical or clinical terminology. In contrast, Clinical BioBERT was pre-trained on domain-specific texts such as PubMed abstracts, PMC full-text articles, and MIMIC-III clinical notes, which differ from the language found in interviews [23]. The observed performance gains from integrating demographic and narrative features in most PLMs may be attributable to the late-fusion strategy, indicating that jointly modeling these inputs can enhance predictive performance. Caregiver burden estimation generally achieved higher performance compared to depression estimation across most models and feature sets, suggesting the risk of caregiver burden may be more readily identifiable from demographic and narrative data.

Feature selection highlighted differences and similarities in attributes associated with the risk of depression versus burden. The consistent selection of *Location* across all folds strongly suggests caregivers' living environments (urban vs. rural) significantly influence these risks, likely reflecting disparities in community resources and support services. Likewise, *Years Caring* was selected in every fold, indicating that caregiving chronicity can be a salient predictor—longer duration may index cumulative stress exposure and sustained role demands. In parallel, the ubiquitous selection of *Income* points to socioeconomic context as a key determinant; limited financial resources can constrain access to respite, healthcare, and supportive services, whereas greater resources may buffer stress. Conversely, *Sex*, *Employment*, and *Health Insurance* were consistently not selected, indicating minimal predictive relevance. Additionally, distinct selection patterns among features such as *Education*, *Age*, *Race*, and *Daily Caregiving Hours* suggest varied underlying factors between depression and burden risks.

Feature importance analysis (Fig. 3) further elucidates key predictors. For depression risk, *Years Caring* and *Income* were primary predictors, underscoring how prolonged caregiving and socio-economic factors might contribute to depressive symptoms. For caregiver burden, *Income*, *Location*, *Daily Caregiving Hours*, *Ethnicity*, and *Race* emerged prominently, emphasizing the influence of caregiving intensity and socio-economic factors on caregivers' perceived burden. *Age* consistently showed minimal influence, likely due to limited variability in our sample's younger age distribution (primarily under 35 years, TABLE I).

## V. CONCLUSIONS AND FUTURE DIRECTIONS

To the best of our knowledge, this study is the first to investigate the feasibility of estimating the risk of depression and caregiver burden among caregivers of PwD by analyzing demographic information and caregiver narratives about caregiving experiences using NLP methods. The trained models demonstrated promising results, achieving F1-scores of 0.74 for estimating the risk of caregiver depression and 0.78 for estimating the risk of caregivers burden on our dataset. These findings highlight the potential of leveraging non-clinical data to facilitate early screening, encouraging caregivers to seek timely medical intervention. These warrant further studies, which can systematically optimize ML algorithms through hyperparameter tuning, exploration of diverse kernels, and other model refinement techniques. Exiting dementia-related studies often predominantly involve white populations (like NACC [29] ADNI [30], ADNIGO/ADNI-2 [31]), resulting in limited representation of African Americans. In contrast, our study primarily includes African American participants, constituting a noteworthy contribution to the existing body of knowledge. However, our findings should be applied carefully as they may not generalize to other populations. Future research should expand the dataset by including caregivers from more diverse demographic backgrounds and caregiving contexts to improve generalizability and performance.

## APPENDIX

1) Think about how you have been feeling in the past few weeks while taking care of your loved one with dementia. Tell us about any recent moments that stand out to you, whether that being positive or challenging.

2) Think about how caregiving has affected your overall well-being in the past few weeks, particularly regarding your relationships with family, friends, or colleagues. Tell us about any changes or shifts that you have noticed in these relationships.

3) Over the past few weeks, have you, as a caregiver, noticed any changes in your decision-making process? If yes, please elaborate.

4) Think about your self-care practices in the past few weeks to manage the daily stresses of caregiving. Tell us your strategies that have been particularly helpful in coping with the daily stresses of caregiving.

5) In the past few weeks, have any factors such as professional commitments, financial constraints, or health changes significantly influenced how you provide care? If yes, please elaborate on these factors and how they have impacted your caregiving responsibilities.

6) As a caregiver, do you sometimes feel concerned about what the future holds for you in that role? If so, could you share more about it? What kind of help or support would be helpful to you as a caregiver?

## ACKNOWLEDGMENT

The authors extend their gratitude to Prof. Seungyeon Lee from the University of Central Arkansas for reviewing the interview probes and providing valuable suggestions.

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
