# OpenReview forum: "Estimating the Risk of Depression and Care  Burden among Dementia Caregivers: A  Feasibility Study"
_IEEE.org/EMBS/BHI/2025/Conference — BHI 2025_

### Official Review · Reviewer_DWUk · 2025-06-29
**Estimating the Risk of Depression and Care Burden among Dementia Caregivers: A Feasibility Study**

**Confidence:** 5
**Clarity Of Writing:** fair
**Clinical Significance:** fair
**Methodological Novelty:** fair
**Overall Rating:** 2
**Final Rating:** 6

**Experiments And Results:**

fair

**Questions For The Authors:**

Thank you for submitting the paper!
Here are a few questions

1. Could you clarify the specific feature selection methods applied to the narrative features (UNI, ADJ, ACS)? How did you reduce dimensionality or prevent overfitting given the large number of features extracted?

2. The methodology mentions concatenating binary-encoded demographic features with the [CLS] token embeddings from the pretrained language models. Explaining this would help readers understand your model architecture better.

3. Why was k=5 chosen for the k-Nearest Neighbors classifier? Did you explore other values or use the elbow method? Clarification would improve understanding of model selection.

4. The discussion notes that combining demographic and narrative features degraded the performance of pretrained language models. Could you elaborate on the possible reasons for this and any experiments you conducted to understand this issue?

5. Could you provide more detail on how missing demographic data was handled and its impact on the models?

**Strengths:**

1. The introduction effectively frames the importance of the problem, clearly explaining the challenges faced by dementia caregivers and the need for early screening of depression and caregiver burden.

2. The study uses a rigorous cross-validation strategy with feature selection performed within each fold, which helps prevent data leakage and overfitting.

3. The multi modal approach of combining demographic data and narrative text is a promising approach, reflecting real-world conversational settings and potentially enabling non-clinical early screening.

4. The effort to compare traditional machine learning models with pre-trained language models adds value by exploring the applicability of modern NLP techniques to the caregiving domain.

**Summary Of The Paper:**

This paper investigates the feasibility of estimating depression risk and caregiver burden among dementia caregivers using demographic data and natural language narratives about caregiving experiences. The paper analyzed data from 65 caregivers through surveys and interviews, and applied a combination of traditional machine learning models and pretrained language models to classify depression and burden risk. They explored multiple feature sets including demographics alone, narrative text alone, and combined features. The study emphasizes the potential of leveraging conversational and demographic information as non-clinical tools to support early identification of caregiver distress.

**Weaknesses:**

1. The study’s primary contribution is limited by the small sample size (N=65), which restricts the generalizability and robustness of the findings. A larger, more diverse dataset is needed to strengthen the conclusions as mentioned in the limitations.

2. Narrative features extracted using basic Bag-of-Words representations (UNI, ADJ, ACS) appear insufficiently informative, with demographic features dominating model performance. This suggests that the narrative data, in its current form, contributes little beyond demographics, which limits the novelty and potential impact of the approach. Exploring more advanced NLP techniques such as sentiment analysis, semantic embeddings could better capture the complexity of caregiver narratives. This may also suggest that the current feature extraction methods do not fully capture the meaningful content of caregiver narratives, limiting the added value of the language data.

3. The paper lacks clarity on the feature selection methods applied to the narrative features, making it difficult to assess how these high-dimensional text features were filtered or reduced before model training, raising concerns regarding overfitting.

4. The choice to exclude popular machine learning models such as Random Forest or Decision Trees, which often handle mixed data types and feature importance well, is unexplained and may represent a missed opportunity.

5. The use of only F1-score as the evaluation metric is limiting; additional metrics such as precision, recall, and ROC-AUC should be reported to better characterize model performance, especially given the clinical relevance of incorrectly classified instances.

6. There is limited discussion of model errors or failure modes, such as analysis of misclassified cases, which would provide valuable insights into the model’s limitations and areas for improvement.

7. The manuscript’s writing quality could be improved for greater clarity and readability.

8. The paper does not clearly state the “why” behind the study beyond general importance. While the introduction outlines the problem well, it lacks a focused explanation of the specific research gap or unmet need that this work addresses, which reduces clarity about the study’s unique contribution.

---

### Official Review · Reviewer_n1SP · 2025-07-11
**Estimating the Risk of Depression and Care Burden among Dementia Caregivers: A Feasibility Study**

**Confidence:** 5
**Clarity Of Writing:** great
**Clinical Significance:** good
**Methodological Novelty:** great
**Overall Rating:** 7

**Experiments And Results:**

good

**Questions For The Authors:**

Everything is clear. I have no questions.

**Strengths:**

This work addresses well-known topics, such as the analysis of depression and caregiver burden among people caring for individuals with dementia (PwD). Classification models and BERT-based architectures are commonly used for similar tasks. However, the promising aspects of this study lie in the combination of demographic and narrative information, and in the use of Bio+Clinical BERT, a domain-specific model pre-trained on biomedical and clinical texts, which is particularly relevant in this context.

**Summary Of The Paper:**

The paper investigates how to assess the risk of depression and caregiver burden in individuals supporting people with dementia. Recognizing that caregivers often experience emotional and physical stress, the authors propose using informal conversations with social workers as a potential screening tool. everal machine learning models were trained, including Logistic Regression, SVM, k-Nearest Neighbors, and fine-tuned versions of BERT and Bio+Clinical BERT. SVM achieved the best overall performance. For depression risk prediction, the highest F1-score (0.74) was obtained by combining demographic data with adjective-based narrative features. For caregiver burden, demographic features alone yielded the best result (F1-score: 0.78).

**Weaknesses:**

In my opinion, the paper does not report any weaknesses

---

### Official Review · Reviewer_7qGD · 2025-07-17
**Estimating the Risk of Depression and Care Burden among Dementia Caregivers: A Feasibility Study**

**Confidence:** 4
**Clarity Of Writing:** good
**Clinical Significance:** good
**Methodological Novelty:** good
**Overall Rating:** 6

**Experiments And Results:**

good

**Questions For The Authors:**

This work makes a valuable interdisciplinary contribution at the intersection of AI, mental health, and caregiving. It addresses a growing public health concern and introduces tools that could inform clinical decision-making and resource allocation in caregiver support systems.

**Strengths:**

Novelty and Contribution: This is a pioneering study using unstructured caregiver speech data for estimating both depression and care burden. Few studies in the literature combine linguistic, demographic, and clinical data in this way.

Practical Relevance: The work offers strong implications for developing early-warning tools for social workers and public health practitioners. Real-world applications are evident.

Dataset Collection: Collecting caregiver interview data labeled with PHQ-8 and ZBI is a valuable and non-trivial contribution.

Modeling & Evaluation: Using Leave-One-Subject-Out Cross-Validation (LOSO) is an appropriate strategy given the small sample size. The inclusion of SHAP analyses for model explainability is commendable.

**Summary Of The Paper:**

This study presents an innovative approach to estimating depression risk and care burden among dementia caregivers by leveraging natural language processing (NLP) and machine learning (ML) models on speech transcripts. Using clinically validated scales (PHQ-8 and ZBI) as ground truth labels, the authors construct a multi-modal dataset and evaluate a wide range of model types and feature sets.

The study is technically sound, timely, and addresses a relevant and underexplored problem space. It holds strong potential for real-world impact, especially in the early identification of at-risk caregivers in social work or clinical settings. However, there are limitations in terms of sample size, generalizability, and some methodological justifications that could be strengthened.

**Weaknesses:**

>The sample size is very small (N = 65), which limits statistical power and may hinder the performance of deep learning models such as BERT.
>The demographic distribution is highly skewed (e.g., ~90% aged 18–34, ~85% African American), limiting the generalizability of the findings.
>The dataset is not publicly available, making replication and further exploration by other researchers difficult.
>The performance drop in PLM (pretrained language model) models needs deeper discussion. For example, why does Bio+Clinical BERT underperform compared to general BERT?
>When combining demographic and language features, some PLMs perform worse — this is observed but not explained.
>Some feature engineering choices (e.g., focusing on adjective-containing sentences) appear somewhat heuristic and could benefit from stronger theoretical or empirical justification.
>The paper is comprehensive but occasionally overwhelming. The Results section in particular would benefit from a clearer summary and more concise tables.